# DisFormer: Disentangled Object Representations for Learning Visual Dynamics Via Transformers

## Abstract

We focus on the task of visual dynamics prediction. Recent work has shown that object-centric representations can greatly help improve the accuracy of learning dynamics. Building on top of this work, we ask the question: would it help to learn disentangled object representations, possibly separating the attributes which contribute to the motion dynamics vs which don't? Though there is some prior work which aims to achieve this, we argue in this paper either it is limiting in their setting, or does not use the learned representation explicitly for predicting visual dynamics, making them sub-optimal. In response, we propose *DisFormer*, an approach for learning disentangled object representation and use them for predicting visual dynamics. Our architecture extends the notion of slots Locatello et al. (2020) to taking attention over individual object representations: each slot learns the representation for a block by attending over different parts of an object, and each block is expressed as a linear combination over a small set of learned concepts. We perform an iterative refinement over these slots to extract a disentangled representation, which is then fed to a transformer architecture to predict the next set of latent object representations. Since our approach is unsupervised, we need to align the output object masks with those extracted from the ground truth image, and we design a novel permutation module to achieve this alignment by learning a canonical ordering. We perform a series of experiments demonstrating that our learned representations help predict future dynamics in the standard setting, where we test on the same environment as training, and in the setting of transfer, where certain object combinations are never seen before. Our method outperforms existing baselines in terms of pixel prediction and deciphering the dynamics, especially in the zero-shot transfer setting where existing approaches fail miserably. Further analysis reveals that our learned representations indeed help with significantly better disentanglement of objects compared to existing techniques.

## 1 Introduction

Predicting visual dynamics is an important problem that finds applications in computer vision, model-based reinforcement learning, and visual question answering. Some of the classic work does this by having dense representation for the image and then passing through a CNN-based or now transformer-based architecture. More recent approaches have argued for the use of object-centric representations, presumably because they can better capture the natural semantics of how objects are represented in a scene and their interactions, for the task of dynamics prediction. This line of work can further be divided into two sets of categories: ones that work with dense object embeddings learned from the data Wu et al. (2023) and those that try to decompose each object representation in terms of its attributes Singh et al. (2022b). While in the former category, the latest models exploit the power of transformers to get the future dynamics, works in the second category have exploited autoencoder-based models, or GNN-based models, to learn disentangled representation for downstream learning of dynamics. Interestingly, some of the SOTA models Wu et al. (2023) do not use disentangled representation, leaving a possibility of further improvement in performance on this task while having a more interpretable representation.

Even among the techniques that make use of a disentangled object representation, they are limited by their assumptions: (1) They either work with objects of the same size Kossen et al. (2019) (2) Or divide the attributes only into two sets: those relevant for dynamics, vs which are not Nakano et al. (2023). Both these assumptions are pretty restrictive. The closest to our work is Lin et al. (2020a), which can learn disentangled representations, but it does it with a fixed number of concepts independent of the problem space. As observed in our experiments, this severely affects their performance, especially in the setting of transfer, for instance, where certain attribute combinations are never seen before. Motivated by these research gaps, we push the boundaries on two fronts: (1) Work with a more flexible disentangled representation, which can choose the number of concepts depending on the specific problem; (2) Combine the learned representation with the power of transformer-based architectures, which have been shown to do very well on future prediction tasks. We refer to our system as *DisFormer*.

Our work, in the form of DisFormer, presents several novel contributions. Starting with masks extracted from an object extractor, our key idea is to represent each object as a set of blocks, where each block is expressed as a linear combination of underlying (learned) concepts. Intuitively, each block (or a set of blocks) can be thought of as potentially representing an underlying natural attribute of an object (e.g., size, colour etc.). While such an idea has been explored in earlier work Singh et al. (2022b), it is limited to the case of static images, whereas our goal is to tie the learned representation with dynamics prediction, resulting in some significant technical differences. In order to learn the individual block representation, we use the notion of a slot Locatello et al. (2020). Whereas in the original slot attention paper, each slot binds to a particular object, and its representation is obtained by iteratively refining a vector attending over the entire image, in our case, each slot represents a block, which attends over an object, and its iterative refinement results in the final representation of the block. Disentanglement is guided by forcing each block to be a linear combination of a small number of learned concepts. The final object representation is the concatenation of the blocks.

For dynamics prediction, we pass the latent representation for the entire set of extracted objects in the image, along with one for the background, to a transformer. A position embedding for each block distinguishes it from others. Since object masks can be discovered in any order, we need to impose a canonical ordering over them so that loss can be taken for the final image. This is done via a novel permutation module, which learns to map object representations to a canonical ordering starting from any given ordering over them. Finally, instead of passing the object representation as a single vector during decoding, we directly map the discovered blocks to individual channels, further reinforcing the discovery of a disentangled representation. For the object extractor, we make use of a Mask-RCNN trained on labels provided by an unsupervised extractor Gupta et al. (2021), for 2D scenes, and slot attention Locatello et al. (2020), for the case of 3D scenes. For our decoder, we use a standard architecture Watters et al. (2019) along with mapping of attribute representation to channels as described above.

Extensive experimentation on both 2D and 3D datasets shows that our approach outperforms existing SOTA baselines both in terms of pixel error and dynamics, for up to 15 steps of prediction. Further, we create transfer datasets for 2D environments and we observe that our model performs much better in zero-shot settings than the baselines that fail miserably[1]. The rest of the paper is organized as follows: in section 2, we will discuss the related work, and in section 3, we will propose our architecture. Section 4 discusses the experiments and results, followed by a conclusion and discussion on future work in section 5.

## 2 RELATED WORK

The task of video prediction can be seen as a combination of image synthesis and learning dynamics of various entities in the scene. There has been a tremendous amount of work in image synthesis using unstructured representations like Variational Autoencoders (Kingma & Welling, 2014), Generative Adversarial Networks (Goodfellow et al., 2014) and their variants (Makhzani et al., 2016; Tolstikhin et al., 2018; Dai & Wipf, 2019; Mondal et al., 2020; 2021; Ghosh et al., 2020; Arjovsky et al., 2017; Gulrajani et al., 2017). For video prediction, the initial works focused on direct pixel synthesis using hybrid architectures (Shi et al., 2015; Wang et al., 2017) that fuse recurrent neural networks and convolutional networks. However, these methods struggled to do long-horizon predic-

---

[1]We will release our code and datasets for further research after acceptance.

tions. Another line of work (Dosovitskiy et al., 2015; Sun et al., 2018; Reda et al., 2018) leverages optical flow information to forecast the movement of objects and pixels within a video sequence. While these methods are relatively interpretable and capture a notion of motion, they still fail to capture any higher level of semantics and object interactions. To have better interpretability and control over the learned representation space, some parts of the research eventually shifted to learning interpretable and structured image and video representations. We will discuss the developments in this area next.

**Object-Centric Approaches:** There has been a growing interest in learning object-centric representations of images in an unsupervised manner by decomposing a scene into its constituting objects. Some of the notable works include CSWM (and variants) Kipf et al. (2020); Gupta et al. (2021), AIR (Eslami et al., 2016), MONet (Burgess et al., 2019), IODINE (Greff et al., 2019), GENESIS (Engelcke et al., 2019), SPACE (Lin et al., 2020b), Slot Attention (Locatello et al., 2020), GENESIS-V2 (Engelcke et al., 2021), SLATE (Singh et al., 2022a), or Neural Systematic Binder (Singh et al., 2022b). Slot Attention defines the notion of *slots* where each slot is tied to an object in the scene, and it learns the object's representation using iterative refinement. Another line of work segments images by taking the last couple of frames as input; these include SAVi (Kipf et al., 2022), SAVi++ (Elsayed et al., 2022) and STEVE (Singh et al., 2022c) extend Slot Attention and VideoSaur (Zadaianchuk et al., 2023) uses self-supervised optical flow based loss.

Motivated by these successes, video prediction approaches were built by utilizing some of these methods: SQAIR (Kosiorek et al., 2018) extends AIR, STOVE Kossen et al. (2019), GSWM Lin et al. (2020a), OCVT Wu et al. (2021) and SlotFormer Wu et al. (2023). STOVE learns a factorized representation per object with explicit position and size of each object; it then uses a Graph Neural Network as the dynamics model. GSWM does explicit factorization of each object whereas we factorize implicitly. OCVT also has limited explicit factorization of each object and uses a transformer for learning dynamics. SlotFormer extends Slot-Attention to video prediction and uses a transformer for dynamics prediction, however, it does not have disentangled object-level representation.

**Object-centric representations for transfer learning:** There has been a growing interest in learning neuro-symbolic systems that can transfer a learned solution to unseen combinations of objects. These methods formulate the representation space into its constituting objects and then train a GNN or transformer, leveraging their size-invariant nature, to learn transferable solutions. Works like Yoon et al. (2023); Sharma et al. (2023b) apply this to learn generalized policy in Reinforcement Learning, Ståhlberg et al. (2022); Sharma et al. (2022; 2023a) represent a state in relational planning as a graph of objects, to learn a generalized policy using a GNN. In contrast, our work learns disentangled object representations that help transfer dynamics to unseen combinations.

## 3 DISFORMER

We describe the architecture of DisFormer in this section. There are five important parts to the architecture: (1) Object Extractor: This module extracts dense object representations in an unsupervised manner using pre-trained architectures. (2) Block Extractor: This a novel module that disentangles each object in terms of an underlying *block* representation via iterative refinement (3) Permutation Module: This is a novel module to enforce permutation invariance across the discovered set of objects. (4) Dynamics Predictor: This module predicts the next state of object latents, in terms of their block representation via a transformer-based architecture (5) Decoder: This module inputs the disentangled object representation and decodes it to get the final image. Figure 1 describes the overall architecture. We next describe each module in detail.

### 3.1 OBJECT EXTRACTOR

We train object extractors independently to extract masks from frames. We freeze the object extractor during subsequent training. All our object extractors are unsupervised and trained in a self-supervised manner. In our experiments, for 2D environments, we train an expert model similar to (Sharma et al., 2023b) to generate supervised data for Mask R-CNN (He et al., 2017). For 3D environments, we train Slot Attention (Locatello et al., 2020) and use decoder masks during training. Our architecture generally allows the flexibility of using any object extractor model.

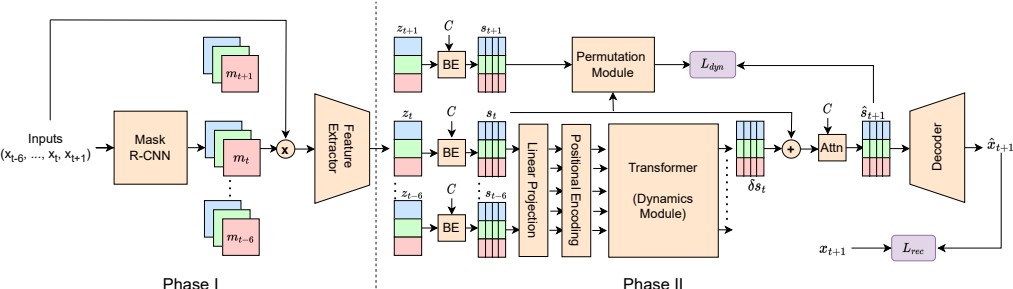

Figure 1: Main architecture for DisFormer. Mask R-CNN in the diagram is the Object Extractor, and the product operation is the Hadamard Product between Object Masks and the input images. For 3D environments, Mask R-CNN would be replaced by Slot Attention. BE represents the Block Extractor module, which takes in the object representations $z_t^i$, one at a time, along with concept vectors $C$ and outputs the set of block representations for each object $i$. Note that each block has its own set of concept vectors; thus, $C = ||_{j=1}^r C_j$ where $r$ represents the number of blocks. Attn is the simple dot-product attention module where the input block-based representation of all the objects is converted into a linear combination of the corresponding concept vectors. Phase I and II represent the two training phases. Other modules are as defined in Section 3

Formally, given a sequence of frames $\{x_t\}_{t=1}^T$, for $T$ time-steps, where $x_t \in \mathbb{R}^{H \times W \times 3}$, we represent the extracted object masks as $m_t^i \in [0,1]^{H \times W}$ where $i \in 1, \cdots, N$ represents the discovered number of objects. These extracted object masks are then multiplied element-wise with the corresponding input frame and passed through a pre-trained feature extractor to obtain latent object representations denoted by $z_t^i \in \mathbb{R}^{f \times d_f}$

## 3.2 BLOCK EXTRACTOR

In our disentangled representation, each object is represented by a set of blocks, and a disjoint set of blocks represents an attribute of an object. We aim to discover these (latent) blocks automatically from the data. Recently, (Locatello et al., 2020) in the paper on slot-attention proposed architecture for unsupervised discovery of objects via iterative refinement of what is referred to as a *slot*. Each slot binds to an object via attention over the entire image in their work. We extend their idea to the case of objects, where each slot now represents a block, which is iteratively refined by taking attention over a latent object representation. Further, to learn a disentangled representation, we enforce that each block representation is represented as a linear combination of a fixed set of underlying learnable concepts. This combination is discovered at every step of the block-refinement algorithm.

More formally, given a latent object representation $z_t^i$ for the $i^{th}$ object, its representation in terms of blocks is given as $\{s_t^{i,b}\}_{b=1}^r$, where $r$ is the number of blocks, and is a hyperparameter of the model. We let each $s_t^{i,a} \in \mathbb{R}^{d_b}$. Algorithm 1 outlines the iterative refinement step to obtain the block representation and closely follows the description in Locatello et al. (2020), with the image features replaced by object features and each slot representing blocks instead of individual objects. Each $s_t^{i,a}$ is initialized to a learnable vector $\mu_b$ to break permutation invariance among blocks [2]. After an initial layer normalization, we make the slot vectors attend over object features to compute the attention scores. These are first normalized over queries (slots), followed by normalization over keys (object features). The resultant linear combination of object features is first passed through a GRU, followed by an MLP to get the block representation. Finally, unique to our approach, we project each resultant block vector onto the learnable concept space and update its representation as a linear combination of the concepts via projection weights (lines 9 - 12 in Algorithm 1). This step results in discovering the disentangled representation central to our approach.

We note that recently Singh et al. (2022b) proposed a similar-looking idea of learning disentangled representation for objects as a linear combination of concepts using an iterative refinement over slots, albeit for static images. Our approach is inspired by their work but has some important

---

[2]Locatello et al. (2020) initialize their slots randomly

differences. In their case, the slots still represent objects as in the original slot-attention paper, making the disentanglement closely tied to the learning of object representations. This also means that their approach is limited by a specific choice of object extractor. In contrast, since we work with object representations directly, our approach is oblivious to the choice of the extractor. Further, as already pointed out, their work is limited to static images, whereas we would like to learn these representations for dynamics prediction.

---

**Algorithm 1** Block Extractor: Inputs are: object features $z_t^i \in \mathbb{R}^{f \times d_f}$. Model parameters are: $W_K \in \mathbb{R}^{d_f \times d}$, $W_Q \in \mathbb{R}^{d_b \times d}$, $W_V \in \mathbb{R}^{d_f \times d_b}$, initial block values $\mu \in \mathbb{R}^{r \times d_b}$, concept vectors $C_b \in \mathbb{R}^{k \times d_b}$ for $b \in \{1, .., r\}$, MLP, GRU

---

1: $s_t^i = \mu$
2: **for** $t = 1$ to $T$ **do**
3:     $s_t^i = \text{LayerNorm}(s_t^i)$
4:     $A = \text{Softmax}((\frac{1}{\sqrt{d}}(s_t^i W_Q)(z_t^i W_K)^T, \text{axis} = \text{'block'})$
5:     $A = A.\text{normalize}(\text{axis='feature'})$
6:     $U = A(z_t^i W_V)$
7:     $U = \text{GRU}(\text{state} = s_t^i, \text{input} = U)$
8:     $U = U + \text{MLP}(U)$
9:     **for** $b = 1$ to $r$ **do**
10:         $w_t^{i,b} = \text{Softmax}(\frac{1}{\sqrt{d_b}} C_b [U[b :]]^T)$
11:         $s_t^{i,b} = C_b^T w_t^{i,b}$
12: **return** $s_t^i$

---

## 3.3 DYNAMICS

Transformers(Vaswani et al., 2017) have been very effective at sequence-to-sequence prediction tasks. Some recent work on unsupervised video dynamics prediction Wu et al. (2023) has given SOTA results on this task on 3-D datasets, and have been shown to outperform more traditional GNN based models for this task Kossen et al. (2019). While some of these models are object-centric, they do not exploit disentanglement over object representation. Our goal in this section is to integrate our disentanglement pipeline described in Section 3.2 with downstream dynamics prediction via transformers. Here are some key ideas that go into developing our transformer for this task:

1. We linearly project each $s_t^{i,b}$ to a $\hat{d}$ dimensional space using a shared $W_{proj}$. $s_t^{i,b} = s_t^{i,b} W_{proj}$, where $W_{proj} \in \mathbb{R}^{d_b \times \hat{d}}$.

2. The input to transformer encoder is $T$ step history of all object blocks that is, at time step $t$, the input is $\{s_{t-\bar{t}}^{i,b} | \bar{t} \in \{0, .., T-1\}, i \in \{1, .., N\}, b \in \{1, .., r\}\}$.

3. We need positional encoding to distinguish between (a) different time steps in the history (b) blocks belonging to different objects (c) different blocks in an attribute. Accordingly, we design 3-D sinusoidal positional encodings $P_{t,i,b}$ and add them to block latents. That is the final vector $s_t^{i,b} = s_t^{i,b} + P_{t,i,b}$.

4. Let the transformer encoder output be $\delta \hat{s}_t^{i,b}$. In principle, it can be directly added to $s_t^{i,b}$ to obtain latent $\hat{s}_{t+1}^{i,b}$ at time step $t+1$. But we rather exploit the fact that object blocks should be linear combination of *concept vectors*. Therefore, we define $\hat{v}_{t+1}^{i,b} = s_t^{i,b} + \delta \hat{s}_t^{i,b}$, and compute $\hat{w}_t^{i,b,j} = \hat{v}_{t+1}^{i,b} \cdot C_{b,j} / ||C_{b,j}||^2$. Then, $\hat{s}_{t+1}^{i,b} = \sum_j \hat{w}_t^{i,b,j} C_{b,j}$.

Finally, we note that unlike (Wu et al., 2023), who have permutation equivariance between object representations, since object slots at any given time step are initialized based on slots at the previous time step, our object latents are obtained from an extractor which may not guarantee any specific ordering among object latents across time steps. Therefore, in order to compute dynamics loss between the actual latents $s_{t+1}^{i,b}$ and predicted latents $\hat{s}_{t+1}^{i,b}$, we need to ensure consistent ordering among object block representations which are input, and the ones which are predicted. We design a novel permutation module to achieve this objective, and we describe it next.

## 3.4 PERMUTATION MODULE

Let $\mathbf{s}_t^i = ||_{b=1}^r s_t^{i,b}$ denote the concatenation of block vectors for object $i$ at time step $t$. Similarly, we define $\hat{\mathbf{s}}_t^i = ||_{b=1}^r \hat{s}_t^{i,b}$. We treat them as column vectors. We note that because $s_t^{i,b}$ is used to predict $\hat{s}_{t+1}^{i,b}$, $\forall b$, in the form of a residual, we expect that the ordering of objects in predicted next slots is same as that of immediate last time step ordering. The key issue is between the ordering of $s_{t+1}^{i,b}$ and $\hat{s}_{t+1}^{i,b}$ which will not be the same in general. We are then looking for a permutation matrix $P \in \mathbb{R}^{N \times N}$ such that $\mathbf{s}_{t+1}^i$ when permuted by $P$ aligns with $\hat{\mathbf{s}}_{t+1}^i$ so that dynamics loss can be computed. Further, we note that when predicting for $t'$ time steps ahead, given history until time step $T$, during autoregressive generation, all the generated $\hat{\mathbf{s}}_{T+t'}^i$'s follow the same ordering that of $\mathbf{s}_T^i$. We exploit this fact to instead align all the input latents $\mathbf{s}_{T+t'}^i$'s with the ordering in latents $\mathbf{s}_T^i$'s.

Let $M \in [0,1]^{N \times N}$ be a score matrix defined for time-steps $T$ and $T + t'$. Given, $\mathbf{s}_T^i$ and $\mathbf{s}_{T+t'}^j$, $[i,j]^{th}$ of the score matrix $M$ is given as: $M[i,j] = \frac{(U\mathbf{s}_T^i)\cdot(U\mathbf{s}_{T+t'}^j)}{\sqrt{rd}}$

We do this for all pairs of objects $(i,j)$ in the two representations to get the full matrix. Similar to (Mena et al., 2018) we compute soft permutation matrix $P \in [0,1]^{N \times N}$ from $M$. Then, $P = Sinkhorn(M)$. The dynamics loss at time step $T + t'$ is then computed after permuting the input latents with $P$ (see Section 3.6).

Here $U \in \mathbb{R}^{d_p \times rd}$ is learnable matrix. This matrix is learned by supplying random permutations of inputs at various time steps $t \leq T$, computing the permutation matrix $P$, and then computing the loss with respect to the true permutation matrix (which is known). Note that this module is only used during training in order to align the predictions for loss in the dynamics module. We discard it during testing.

## 3.5 DECODER

Same as most of the previous work we use spatial mixture models to generate the final image. As each object is represented by $r$ vectors of blocks, we use a slightly different approach than previous methods which have a single vector representing the object. We first use each block specific Spatial Broadcast decoder (Watters et al., 2019) to generate 2D maps $q_t^{i,b} \in \mathbb{R}^{f' \times I \times I}$ representing $f'$ features each of size $I \times I$, corresponding to $s_t^{i,b}$. We concatenate these blocks to form $\mathbf{q}_t^i \in \mathbb{R}^{rf' \times I \times I}$. A CNN is applied on $\mathbf{q}_t^i$ to generate final object mask which is normalized across the objects $\hat{m}_t^i$ and object content $\hat{c}_t^i$. Final image is obtained as $\hat{x}_t = \sum_{i=1}^N \hat{m}_t^i \cdot \hat{c}_t^i$.

## 3.6 TRAINING AND LOSS

**Curriculum:** We use a history of length $T$ and do a $T'$ step future prediction. We use two-phase training. First Phase: we only train object extractor with object extractor-specific optimization objective. Second Phase: we freeze the object extractor and train the rest of the model. For the first few epochs of training in the second phase, we only train block extractor, permutation module and decoder. Then we only train the dynamic model for a few epochs freezing all other modules. Finally, all modules are trained together except the object extractor.

**Losses:** We make use of the following losses: (a) Image reconstruction loss: $\mathcal{L}_{rec} = \sum_{t=1}^T (\hat{x}_t - x_t)^2$. Captures the overall reconstruction loss. (b) Permutation Loss: $\mathcal{L}_{per} = \sum_{t=1}^{T+T'} (\pi_t - P_t)^2$, where $\pi_t$ is a matrix capturing a random permutation of $\mathbf{s}_t^i$'s, and $P_t$ is the permutation matrix output by the permutation module when input with $\mathbf{s}_t^i$'s and $\pi_t(\mathbf{s}_t^i)$'s. (c) Mask Loss: $\mathcal{L}_{mask} = \sum_{t=1}^T \sum_i (m_t^i - \hat{m}_t^i)$. Captures the loss over predicted masks. (d) Orthogonality Loss: $\mathcal{L}_{ort} = \sum_{b=1}^r \sum_{i,j:i \neq j}^{k,k} |(C_{b,i})^T C_{b,j} / ||C_{b,i}||_2 ||C_{b,j}||_2$. Captures that vectors within each concept should be orthogonal to each other. (e) Dynamic Loss: $\mathcal{L}_{dyn} = \sum_{t=T+1}^{T+T'} \sum_{i,a,j} (\hat{w}_t^{i,b,j} - \tilde{w}_t^{i,b,j})^2$, where $\tilde{w}_t^i$'s represent objects permuted by permutation matrix $P_t$ at time step $t$.

The total loss in Phase II of training is given as: $\mathcal{L} = \mathcal{L}_{dyn} + \mathcal{L}_1$, where $\mathcal{L}_1 = \lambda_{dec}\mathcal{L}_{dec} + \lambda_{per}\mathcal{L}_{per} + \lambda_{mask}\mathcal{L}_{mask} + \lambda_{diss}\mathcal{L}_{diss}$.

# 4 EXPERIMENTS

We perform a series of experiments to answer the following questions: (1) Does DisFormer result in better visual predictions than the existing SOTA baselines on the standard datasets for this problem? (2) Does learning disentangled representations with DisFormer lead to better performance in the zero-shot transfer setting, *i.e.*, when tested on unseen combinations of objects? and (3) Can DisFormer indeed discover the disentangled representation corresponding to natural object features such as color, size, shape, position, etc.? We first describe our experimental set-up, including the details of our datasets and experimental methodology, followed by our experimental results, answering each of the above questions in turn.

## 4.1 EXPERIMENTAL SETUP

### 4.1.1 DATASETS

We experiment on a total of three datasets; two are 2-dimensional and one is 3-dimensional.

**2D Bouncing Circles (2D-BC):** Adapted from the bouncing balls INTERACTION environment in Lin et al. (2020a) with modifications in the number and size of the balls. Our environment comprises three circles of the same size but different colors that move freely in the 2D space with a black background, colliding elastically with the frame walls and each other. Similar environments have been used in STOVE Kossen et al. (2019).

**2D Bouncing Shapes (2D-BS):** We create another dataset, which is an extension of the 2D-BC environment with increased visual and dynamics complexity. Two circles and two squares move freely in the 2D space with a checkered pattern as background. Collisions happen elastically among various objects and frame walls while also respecting respective object geometries. We use the MuJoCo physics engine Todorov et al. (2012) to simulate the domain with camera at the top, and objects with minimum height to have a 2-D environment.

**OBJ3D**: A 3D environment used in GSWM Lin et al. (2020a) and SlotFormer Wu et al. (2023), where a typical video has a sphere that enters the frame and collides with other still objects.

### 4.1.2 BASELINES, METRICS AND EXPERIMENTAL METHODOLOGY

**Baselines:** For the 2D domains, we compare DisFormer with three baselines, STOVE Kossen et al. (2019), GSWM Lin et al. (2020a) and SlotFormer Wu et al. (2023). For the 3D domain, SlotFormer beats both STOVE and GSWM, so we compare with only SlotFormer Wu et al. (2023) in this case.

**Evaluation Metrics:** For 2D-BC and 2D-BS domains, we evaluate the predicted future frames quantitatively using two metrics: (1) Position error: Error in predicted object positions in comparison to the ground truth from the simulator, and (2) Pixel error: MSE Loss between the predicted frame and the ground-truth frame. For OBJ3D, the predicted future frames are evaluated on the PSNR, SSIM and LPIPS metrics as used by Wu et al. (2023).

**Experimental Methodology:** We use the official implementation for each of our baselines: [3]. All the models are given access to 6 frames of history and are unrolled up to 15 future steps while testing. All result table numbers are summed over 15 step future prediction. Each model trains on 1000 episodes, each of length 100, for 2 million training steps on a single NVIDIA A100 GPU, with the exception of SlotFormer which requires two A100's.

## 4.2 VISUAL DYNAMICS PREDICTION

Given a set of past frames, the goal is to predict the next set of frames. Table 1 presents the results on the 2-D datasets. DisFormer beats all other approaches on both pixel-error and pos-error metrics. Our closest competition is GSWM, which does marginally worse on 2D-BS and up to 2.5% worse on 2D-BC in pixel error. In terms of position error, it is 1.5% and 10% worse on the two datasets respectively. STOVE failed to give any meaningful results on 2D-BC dataset. Interestingly, on the 2-D datasets, SlotFormer which is SOTA model on 3D, does worse than some of the earlier approaches

---

[3]STOVE: https://github.com/jlko/STOVE, GSWM: https://github.com/zhixuan-lin/G-SWM, SlotFormer: https://github.com/pairlab/SlotFormer

on the 2-D datasets. We do not report position error for SlotFormer since it does not explicitly work with object masks (and rather only dense representation in terms of slots), and it is not clear how to extract object positions from this representation.

Table 2 presents the results on the OBJ3D dataset. Our numbers are comparable to SlotFormer in this case, with marginal improvement PSNR and marginal loss in LPIPS. This comparable performance in 3D dataset comes at the advantage of being able to disentangle object representations (see Section 4.4).

Finally, we did a small ablation, where on 2D-BC dataset, where we created a variation of our model (called DenFormer, short for Dense-DisFormer) by replacing the block extractor (refer Section 3) by an MLP to create dense object representations. Table 3 presents the results where we have compared with DisFormer, and SlotFormer, which is the only baseline that works with a purely dense object-centric representation. We see that while DenFormer's performance drops, it is still comparable to SlotFormer in pixel error. This only highlights the power of disentanglement but also that in the absence of it, our model becomes comparable to some of the existing baselines.

| Model | 2D-BC | | 2D-BS | |
|---|---|---|---|---|
| | Pixel Err | Pos Err | Pixel Err | Pos Err |
| STOVE | 0.294 | 0.421 | - | - |
| GSWM | 0.283 | 0.417 | 0.7 | 0.711 |
| Slotformer | 0.329 | - | 0.6257 | - |
| DisFormer | **0.28** | **0.402** | **0.608** | **0.610** |

Table 1: Results on 2D Datasets

| Model | OBJ3D | | |
|---|---|---|---|
| | PSNR ($\uparrow$) | SSIM ($\uparrow$) | LPIPS ($\downarrow$) |
| Slotformer | 32.40 | **0.91** | **0.08** |
| DisFormer | **32.93** | **0.91** | 0.09 |

Table 2: Results on OBJ3D Dataset

| Model | 2D-BC | |
|---|---|---|
| | Pixel Err | Pos Err |
| Slotformer | 0.329 | - |
| DenFormer | 0.332 | 0.591 |
| DisFormer | **0.28** | **0.402** |

Table 3: Dense vs Disentangled

| Model | 2D-BC-2L1S | | 2D-BS-2L2S | |
|---|---|---|---|---|
| | Pixel Err | Pos Err | Pixel Err | Pos Err |
| GSWM | 0.345 | 0.642 | 0.812 | 0.789 |
| Slotformer | 0.357 | - | 0.794 | - |
| DisFormer | **0.301** | **0.441** | **0.658** | **0.622** |

Table 4: Transfer Setting

### 4.3 TRANSFER LEARNING

**Set-up:** For each of the 2-D datasets in our experiments, we created a transfer learning set-up as follows.
**2D-BC**: We create two variants of 2D-BC in the training dataset, one having larger sized balls and other with relatively smaller balls. The transfer/evaluation then happens on an environment that contains a combination of large and small sized balls.
**2D-BS**: We create a two variants of 2D-BS in a similar manner with different sized objects. The transfer/evaluation then happens on an environment that contains a combination of large and small sized objects.
Table 4 presents the results. Note that, L and S in the environment names represent number of large and small sized objects in the environment. In this case, we clearly outperform all existing baselines on all the metric. This demonstrates the power of learned disentangled representation, which can work seamlessly with unseen object combinations, whereas the performance of other baselines degrades significantly.

### 4.4 DISENTANGLEMENT OF OBJECT PROPERTIES

**Experiment Details:** We conduct a post-hoc analysis to gain insights into the learned disentangled representations of objects in terms of their correlation to their visual features. However, with the latent representations, there is no directly visible mapping from the representations to the objects in the scene. To overcome this challenge, given the trained model and a scene, we perform a forward pass through DisFormer to obtain the block representations for the objects in the scene. Upon

receiving these, we manually perform swapping of different combinations of block representations between the objects (same blocks for different objects) to create the mapping and identify set of blocks that together represent visual features like position, colour and shape etc.

Figure 2 presents the results on the 2-D datasets. In each of the sub-figures, the first two rows represent the masks and contents of the original scene, and the bottom two rows represent mask and content after attributing swapping, respectively. The first column represents the generated image and further columns represent mask and content of decoded slots. In the figure 2a, we swap a single block of magenta circle and red square. The swapped representations after decoding shows that the objects have same position (seen in mask) but swapped the color and shape. Thus that block represents color and shape of an object. In the figure 2b, we swap the two other blocks of magenta circle and green square. The resultant decoded masks and content after swapping shows that objects have swapped the positions (seen in mask) but have retained the same color and shape in content. This shows that the two blocks combined represents the position of an object.

Figure 2c shows the results for swapping two blocks corresponding to red and blue circle from a seen of 2D-BC dataset which results in the positions

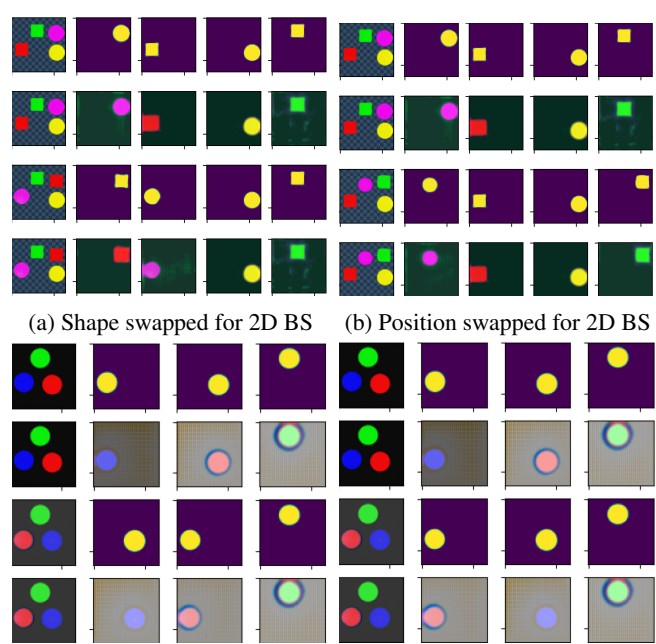

(a) Shape swapped for 2D BS    (b) Position swapped for 2D BS

(c) Position swapped for 2D-BC    (d) Color swapped for 2D-BC

Figure 2: Qualitative Results for disentanglement of object properties on 2D domains. For (a) and (b), attributes of magenta circle have been swapped with corresponding attributes of red and green square respectively. For (c) and (d), attributes of red and blue circles have been swapped.

swapped for two objects as seen in the masks. Thus the two blocks combined repesents position of objects. Similarly after swapping other single block it can be seen in figure 2d that particular block represents the color of objects.

## 5    CONCLUSION AND FUTURE WORK

We have presented an approach for learning disentangled object representation for the task of predicting visual dynamics via transformers. Our approach makes use of unsupervised object extractors, followed by learning disentangled representation by expressing dense object representation as a linear combination of learnable concept vectors. These disentangled representations are passed through a transformer to obtain future predictions. Experiments on three different datasets show that our approach performs better than existing baselines, especially in the setting of transfer. We also show that our model can indeed learn disentangled representation. Future work includes learning with more complex backgrounds, extending to more complex 3D scenes, and extending the work to action-conditioned video prediction.

***Reproducibility Statement.***    To ensure that the proposed work is reproducible, we have included an Algorithm (Refer to Algorithm 1). We have clearly defined the loss functions in Section 3. The implementation details and hyperparameters are specified in appendix 5. The code of the proposed method and datasets will be released post acceptance.

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

# APPENDIX

## A HYPERPARAMETERS

|  |  | BC | BS | OBJ3D |
|---|---|---|---|---|
| Training | Training Steps Phase 1 | 30K | 30K | 60K |
|  | Training Steps Phase 2 | 250K | 250K | 300K |
|  | LR warm-up steps | 1K | 1K | 1K |
|  | Cosine Anneal $T_{cycle}$ | 500K | 500K | 600K |
|  | Cosine Anneal $T_{mult}$ | 1 | 1 | 1 |
|  | Batch size | 32 | 32 | 32 |
|  | $\lambda_{dec}$ | 1 | 1 | 1 |
|  | $\lambda_{per}$ | 10 | 10 | 10 |
|  | $\lambda_{mask}$ | 0.1 | 0.1 | 0.1 |
|  | $\lambda_{diss}$ | 5 | 5 | 5 |
| Block Extractor | Blocks $r$ | 4 | 5 | 8 |
|  | Concepts $k$ | 4 | 4 | 8 |
|  | Block Dimension $d_b$ | 32 | 32 | 64 |
|  | Hidden Dimension $d$ | 64 | 64 | 128 |
|  | $T$ | 3 | 3 | 3 |
|  | Input Dimension $d_f$ | 64 | 64 | 128 |
| Transition Transformer | Dimension $\hat{d}$ | 128 | 128 | 128 |
|  | Layers $k$ | 4 | 4 | 4 |
|  | Heads | 8 | 8 | 8 |
|  | Dropout $d$ | 0.1 | 0.1 | 0.1 |

Table 5: Hyperparameters

| Layers | Stride | Padding | Channels | Activation |
|---|---|---|---|---|
| Conv 3x3 | 1 | 1 | 16 | ReLU |
| MaxPool 3x3 | 2 | 1 | - | - |
| Conv 3x3 | 1 | 1 | 16 | ReLU |
| MaxPool 3x3 | 2 | 1 | - | - |
| Conv 3x3 | 1 | 1 | 32 | ReLU |
| MaxPool 3x3 | 2 | 1 | - | - |
| Conv 3x3 | 1 | 1 | 32 | ReLU |
| MaxPool 3x3 | 2 | 1 | - | - |
| Conv 3x3 | 1 | 1 | 64 | ReLU |
| MaxPool 3x3 | 2 | 1 | - | - |
| Conv 3x3 | 1 | 1 | 64 | ReLU |
| MaxPool 3x3 | 2 | 1 | - | - |

Table 6: Encoder CNN for BC, BS and OBJ3D

| Layers | Stride | Padding | Output Padding | Channels | Activation |
|---|---|---|---|---|---|
| ConvTranspose 5x5 | 2 | 2 | 1 | 32 | ReLU |
| ConvTranspose 5x5 | 2 | 2 | 1 | 32 | ReLU |
| Concatenate the block level output channelwise | | | | | |
| ConvTranspose 5x5 | 2 | 2 | 1 | 32 | ReLU |
| ConvTranspose 5x5 | 2 | 2 | 1 | 32 | ReLU |
| ConvTranspose 5x5 | 1 | 1 | 0 | 5 | ReLU |
| ConvTranspose 3x3 | 1 | 1 | 0 | 4 | ReLU |

Table 7: Decoder for BC, BS and OBJ3D

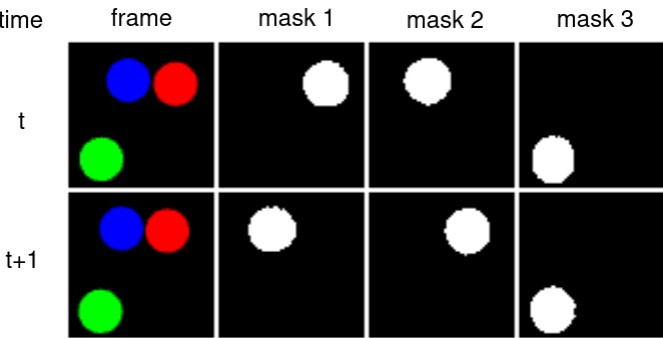

Figure 3: Unaligned output of mask extractor

## B OBJECT EXTRACTOR

We train object extractor beforehand and freeze them during final training. We use two different mask extractor for 2D and 3D environments.

### B1 2D ENVIRONMENT

As we want to perform the transfer experiment, the requirement for mask extractor is that we want it to work on unseen composition of objects. We found that state of the art unsupervised object extractor (Locatello et al. (2020), Lin et al. (2020b), Singh et al. (2022a)) does not give satisfactory results. We found that MaskRCNN provides the compositional generalization we need when trained with labeled data. To generate labeled data in unsupervised fashion we used expert models similar to Sharma et al. (2023b), which are trained on specific variants of dataset. For each variant of dataset we have one expert model. Expert model trained on one varient of dataset does not generalize to other variant of dataset. We generate combined labeled data by using all expert models which is used to train MaskRCNN.

### B2 3D ENVIRONMENT

For 3D environment, we focused on video prediction task for in distribution test data and relaxed the compositional generalization requirement of mask extractor. We used slot attention Locatello et al. (2020) as mask extractor which was trained in unsupervised fashion.

## C PERMUTATION MODULE

Since the order of masks generated by mask extractor is not same (Fig 3) across the video frames we cannot directly use the loss $\mathcal{L}_{dyn} = \sum_{t=T+1}^{T+T'} \sum_{i,a,j} (\hat{w}_t^{i,b,j} - \tilde{w}_t^{i,b,j})^2$. This is because order of objects in $\hat{w}$ and $\tilde{w}$ may be different. We permute the objects in $\hat{w}$ such that they are aligned with objects in $\tilde{w}$. This is achieved by permutation module which returns required permutation matrix using $\hat{s}_t$ and $s_T$. To train permutation module we permute $s_t$ by randomly generated permutation matrix $\pi_t$ and use $s_t, \pi_t(s_t), \pi_t$ as supervised data. We found that even though the permutation module was trained on same time step object representations, it produces correct permutation matrix even for 10 time step apart objects.

## D POSITION CALCULATION

GSWM and STOVE have $z_{pos}$ latent for each object which corresponds to predicted position of object in frame. This $z_{pos}$ was used to compute the MSE for these models. For DisFormer the decoder masks were used to obtain the predicted positions as $x_k^t = \frac{\sum_{i,j} i\hat{m}_k^t[i,j]}{\sum_{i,j} \hat{m}_k^t[i,j]}$ and $y_k^t = \frac{\sum_{i,j} j\hat{m}_k^t[i,j]}{\sum_{i,j} \hat{m}_k^t[i,j]}$

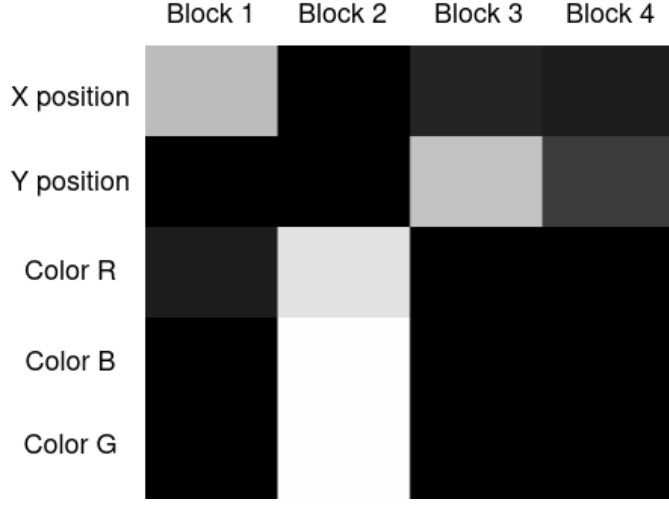

Figure 4: Feature importance mask between blocks and object attributes for BC dataset

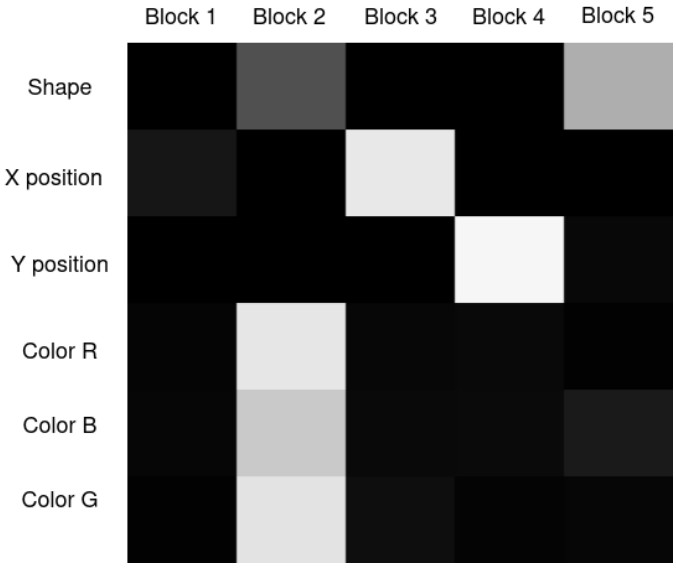

Figure 5: Feature importance mask between blocks and object attributes for BS dataset

## E QUANTITATIVE EVALUATION OF DISENTANGLEMENT

We followed the approach from Singh et al. (2022b) to evaluate disentanglement of blocks. Specifically to visualize and quantify the disentanglement by looking as importance matrix $R \in \mathbb{R}^{A \times r}$ in between the attributes and blocks. To obtain importance matrix, first step is to gradient boosted trees one for each attribute to predict attribute given concatenated blocks of object. After that feature importance vector is generated using permutation importance base. We obtain the importance score of block by adding the importance score of dimensions of blocks.

