# OpenReview forum: "DisFormer: Disentangled Object Representations for Learning Visual Dynamics Via Transformers"
_ICLR.cc/2024/Conference — Submitted to ICLR 2024_

### Official Review · Reviewer_M2hR · 2023-10-30

**Soundness:** 2 fair
**Presentation:** 3 good
**Contribution:** 2 fair
**Rating:** 3
**Confidence:** 3

**Summary:**

The paper proposes DisFormer, which extends Slot Attention and SlotFormer with “disentangled” object representation for visual dynamics prediction. The disentangled representation is learnt by iteratively refining the slots over individual object representations (rather than the whole image representations), and regularizing the slots to be linear combinations of the concepts. To align the model output with the groundtruth, a permutation module is used to learn the ordering. Experiments are done on 2D Bouncing Shapes/Circles and OBJ3D to show the method works well in both in-domain and domain transfer settings.

**Strengths:**

1. The method is clearly motivated and clearly described. By learning slots representing blocks (which are iteratively updated to recover an object representation), and regularizing the slots/blocks to be a linear combination of concepts, the slots/blocks learn disentangled representations of objects.
2. Related works, as well as their difference with this work, are well-discussed.

**Weaknesses:**

My major concern is that the experiments are not very persuasive.
1. The datasets are simple toy datasets. The original SlotFormer has done experiments on dataset CLEVRER, which is harder than the 2D shapes/circles used in this paper. Are there reasons why CLEVRER, or CLEVRER with more complex textures, are used for experiments?
2. No ablations are provided. The model contains multiple components, but there are no ablation experiments to study the effect of each component. For example, the effect of recovering object representation versus image representation, the effect of learning the slots to be a linear combination of concepts, the number of slots, number of concepts, etc. should be studied.
3. Not enough experiments are shown to prove the representations are “disentangled”. This disentanglement is the major advantage of the method. However, only several examples are shown in Fig. 2 to show the disentanglement. Quantitative results, or visualization of the learned slots (e.g. using t-SNE) would be preferred.
4. Missing experiment details. The training details including the hyperparameters are not provided. Some critical parameters (e.g. number of slots/concepts, loss weights) should be discussed.

**Questions:**

See weakness. More details about experiments would be helpful.

---

> ### Author Response · Authors · 2023-11-23
>
> Weakness:
>
> 1. We used OBJ3D which has similar visual complexity of CLEVRER. We are still working to compile our numbers on CLEVRER and hope to shape them during discussion phase.
>
> 2. We are working on specific ablations and hope to shape them during discussion phase.
>
> 3. We have provided disentanglement evaluation in appendix.
>
> 4. Added in the appendix.

---

### Official Review · Reviewer_HX9L · 2023-10-31

**Soundness:** 3 good
**Presentation:** 2 fair
**Contribution:** 3 good
**Rating:** 5
**Confidence:** 4

**Summary:**

This manuscript introduces a novel approach to disentangled object-centric representation learning specifically tailored for video data. The authors present a method that distinguishes itself from prior work, particularly SysBinder, emphasizing its unique applicability to video data and asserting its capabilities in capturing essential knowledge for future prediction in a disentangled manner. Inspired by Slot-Attention, the method utilizes slot-attention on object-centric representations to delineate attributes per the object-centric representation. The model undergoes extensive evaluation across two 2D datasets and a 3D dataset, demonstrating superior performance in pixel-level reconstruction and position estimation over existing methods. Additionally, the authors subject their model to generalization tests in unseen environments, where it continues to outshine competing approaches.

**Strengths:**

- The motivation behind the study is intriguing and thought-provoking.
- The exploration of disentangled object-centric representation for video data is innovative, unveiling new insights, particularly regarding the representation of essential attributes for future prediction from video data.
- The model is intuitively designed, effectively leveraging slot-attention on top of the object-centric representations to disentangle representations.
- Comprehensive comparative analysis, including ablation studies, robustly demonstrates the advantages of disentangled representation.
- The figure for their model architecture is well-drawn and easy to understand.
- The abstract, introduction and the proposed model section are well written except the permutation module section.

**Weaknesses:**

- The manuscript’s clarity and organization can be improved. Specific suggestions for improvement are provided in the questions section.
- A comparative evaluation with another disentangled object-centric representation learning method, SysBinder, is lacking, despite its mention in the text.
- The visualization of disentanglement in the manuscript (Section 4.4) could be enhanced for better clarity and comprehension. Additionally, as they started this paper with the question, “would it help to learn disentangled object representations, possibly separating the attributes which contribute to the motion dynamics vs which don’t?”, if they can show the disentangled representation for the attribute which to contribute to predict the dynamic, it should be much better.
- The quantitative results presented could be bolstered with more illustrative examples, showcasing scenarios where the proposed model excels in comparison to its counterparts.

**Questions:**

### Clarity and Presentation

- In the introduction, could you incorporate a high-level architectural diagram or illustration of your model? This addition would facilitate a clearer and more immediate understanding for readers.
- You’ve described your permutation module as novel. Can you elaborate on its novelty, especially in the context of other existing methods, such as the approach used in the OCVT paper?
- The Mask R-CNN in your methodology is trained using a labeled dataset. This seems to introduce a discrepancy since the other models under comparison do not utilize labeled data. Could you perhaps validate your model's performance using Slot-Attention or another slot-based model as a substitute for Mask R-CNN?
- In Figure 1, “Note that each block has its own set of concept vectors; thus, C = || rj=1C j where r represents the number of blocks.”. Why? Shouldn’t the input $mathcal{C}$ be shared for every block?
- In section 3.2, “we project each resultant block vector onto the learnable concept space and update its representation as a linear combination of the concepts via projection weights (lines 9 - 12 in Algorithm 1). This step results in discovering the disentangled representation central to our approach.”. Could you provide a more in-depth explanation or empirical evidence to support this assertion?
- In Algorithm 1, what is $k$? In figure 1, C is consisted of $r$ concept vectors.
- In section 3.3, “Let the transformer encoder output be δˆ s ti,b. “ For clarity, could you specify that this is the output corresponding to $s_t^{i,b}$?
- Section 3.4 appears to be complex. To ensure my understanding is correct: is the process essentially projecting the concatenated block vectors through matrix $U$, calculating the Cosine Similarity, and then aligning the most similar representations as the same object? Further clarification and polishing of this section would be beneficial.
- How is the object position estimated within your model, as detailed in the experiment section? You've mentioned that position error is not reported for SlotFormer due to its lack of explicit object mask handling; does your model operate in a similar manner?
- Could you provide a deeper analysis of DenseFormer, particularly in comparison to Slotformer? Are there specific scenarios where DenseFormer is more prone to failure, and could you share sample outputs from both DisFormer and DenseFormer to illustrate these points?
- In Section 4.3, the term “transfer learning” is used. Based on my understanding, the experiments seem to be more about evaluating generalization to unseen environments rather than transfer learning. Would renaming this as generalization and providing a more comprehensive analysis, especially in light of the varied performance across different datasets, be more accurate and informative? For example, when comparing the results in Table 1, Slotformer performance deterioration for 2D-BC is not huge while Slotformer is worse for 2D-BS. This results can suggest that for more complicated environment, the disentangled representation is more helpful for the generalization.
- The examples in Section 4.4 intended to illustrate disentanglement seem to be lacking. Instead of swapping both color and shape attributes, could separate demonstrations of each be more effective in showcasing the model’s capabilities?

### Methodology
- Regarding Section 3.4, why is the permutation module utilized only during training and discarded during testing? Could its application during testing, potentially for aligning the input order of the Transformer module, lead to enhanced performance?
- For the Dynamic loss calculation in Section 3.6, why is the comparison made between the weights of the concept vectors rather than the block vectors? Additionally, could you provide a definition for $\hat{w}_t^{i,b,j}$?

### Experiment
- In Section 3.6, there is a training phase where only the block extractor, permutation module, and decoder are trained, followed by a phase focusing solely on the dynamic model. Could you elaborate on the reasons and potential benefits of this training strategy? Did it result in improved model performance?
- The model training in Section 3.6 incorporates multiple loss functions. Have ablation studies been conducted to understand the impact of each loss function on the overall performance?

### Additional Comments
The study presents an interesting investigation with noteworthy contributions to the field. However, to ensure a stronger impact and facilitate better understanding, a revision focusing on improving presentation, clarity, the comparison with the relevant work, and depth of analysis are recommended.

---

> ### Author Response · Authors · 2023-11-23
>
> 1. We would like to point out to reviewer that Sysbinder discovers object representation for images and we would like to discover object representations for video. We still performed used frame by frame sysbinder and found satisfactory results on 2D BS dataset.
> 2. We are working on compiling example images and hope to post them in the discussion phase.
> 3. We are working on high level diagram and hope to post them in the discussion phase.

---

> ### Author Response · Authors · 2023-11-23
>
> 1. We are working on high level diagram and hope to post them in the discussion phase.
>
> 2. OCVT’s alignment algorithm is based on hungarian matching algorithm over L2 norm of latents. In our experiment it performed poorly. We hope to add some results during discussion phase.
> 3. We have added the details in the appendix section
> 4. Each block has it’s own set of concept vectors. For instance the block(s) representing color would have concept vectors as {red, green, blue} etc. whereas those representing size would have a different set of concept vectors. two different attributes cannot be a linear combination of same concept vectors. Thus concept C is not shared across all the blocks.
> 5. k is number of concept vectors
> 6. Yes it is output corresponding to $s_{t}^{i,b}$
> 7. Yes the reviewers understanding is correct.
> 8. We have added position error computation section in appendix
> 9. In our choice of dataset the color and shape are correlated and are not represented by separate blocks.

---

> ### Author Response · Authors · 2023-11-23
> **Methodology**
>
> 1. That's very interesting suggestion. We are currently working on it and hope to add results in discussion phase.
> 2. We found that comparing weights of concept vectors result in faster convergence. The $\hat{w}_{t}^{i,b,j}$ is the predicted weight of $b^{th}$ block and its jth concept vector. It is output of transition model.

---

### Official Review · Reviewer_gVGK · 2023-10-31

**Soundness:** 3 good
**Presentation:** 2 fair
**Contribution:** 2 fair
**Rating:** 3
**Confidence:** 4

**Summary:**

In this work, a novel model is proposed for next frame prediction for videos of interacting objects. Building on previous architectures, the authors explore
further structuring an object-centric representation into blocks that represent
specific object attributes such as shape or color. Object-centric representations are
first obtained using a pretrained unsupervised segmentation model and a feature
extractor. Slot Attention is used to decompose object representations into blocks. A
Transformer than predicts the latent representation of the next frame which is converted into an image using an adapted Spatial Broadcast Decoder. The model consistently
outperforms previous models on three synthetic datasets.

**Strengths:**

- Learning scene representations that are structured into objects and their attributes
  is a very relevant topic. An unsupervised approach based on next frame prediction as
  followed by the authors is applicable in a broad range of settings.
- The proposed model consistently improves over previous methods.

**Weaknesses:**

The empirical evaluation has a strong focus on measuring quantiative performance
averaged over entire datasets. Further insights into the inner workings of the model or
components necessary for outperforming previous approaches are hardly provided.
- This paper proposes a range of novel model components, a detailed ablation analysis is
  however missing. The only comparison is to a model that replaces the pretrained object extractor with an MLP. So it is not clear to which degree the different components
  contribute to the improved performance of the model.
- The paper does not discuss any particular success or failure cases of the proposed
  model. Are there specific situations which are predicted better by the proposed model? How do these related to the model components introduced in the paper?
- The model learns a constant concept space $C$. Do the concept vectors correspond to
  interpretable attributes? Is there a separation of object attributes into those that
  contribute to dynamics and others that do not, as asked in the abstract?
- The disentanglement of object attributes is a core motivation behind this work. The
  disentanglement is however not quantitatively evaluated.

**Questions:**

- In Phase I of the model, object masks are predicted and the masked objects passed
  through a feature extractor. Why is this extra step necessary? Is it possible to use
  the internal representation of the segmentation module directly?
- In section 3.2: How exactly are block vectors projected onto the learnable concept
  space? A mathematical description might be helpful here. Why does this project lead to
  disentangled representations?
- How were hyperparameters tuned for the DisFormer? How sensitive is the model with
  regard to chosing hyperparameters?
- Were the hyperparameters tuned for the baselines?
- How are the predicted object positions obtained for all methods when evaluating the
  position error?
- I do not understand the transfer learning setup in section 4.3: Two variants of the
  training set are created, are the models trained on both? How is it different from the
  evaluation setting?

---

> ### Author Response · Authors · 2023-11-23
>
> 1. Most of our model components are dependent on each other, so the entire system may not work if we remove any of those. At the same time, if the reviewer has specific suggestions on which ablations to try with respect to moving out certain components, we will be happy to report those. There was a typo in experiment section 4.2 where instead of object extractor we replaced block extractor with MLP.
>
> 2. This is a very important suggestion. We are working on compiling a list of failure cases of our model. We hope to upload this to these before the end of the discussion phase.
>
> 3. In our ablation experiment, we rigidly assigned concept weights for blocks to values of 0 and 1. Unfortunately, the decoding process with these blocks failed to produce any meaningful images. This experiment has conclusively shown that, under these conditions, individual concept vectors do not effectively represent interpretable attributes. About the second part of the question we are working defining these sepration. We hope to share the results before the end of the discussion phase.
>
> 4. We have added the appendix section of disentanglement evaluation.

---

> ### Author Response · Authors · 2023-11-23
>
> 1. Internal representation of mask does not have appearance information of object. We thank the reviewer for the suggestion. We would also like to point out that we also need the appearance information of the object and only the mask's hidden representation will not be sufficient. We are working on the reviewers suggestion and hope to share results in the discussion phase.
>
> 2. The line 10 and 11 in Algorithm 1 does the projection of blocks to concepts. As the concept vectors for a given block common across the objects these add strong prior for learning disentangled representation.
>
> 3. We performed a grid search of hyperparameters for a reasonable range. We found that DisFormer’s training is stable in the range of consideration.
>
> 4.  We started with hyperparameters reported by the baselines and tuned in limited space.
>
> 5. We added a section in appendix describing this.
>
> 6. In section 4.3 single model was trained on two datasets. The first dataset have all small objects and second dataset have all big objects in video. During testing we test on a dataset where some objects are small and some are big.

---

### Official Review · Reviewer_987R · 2023-11-01

**Soundness:** 2 fair
**Presentation:** 2 fair
**Contribution:** 2 fair
**Rating:** 3
**Confidence:** 3

**Summary:**

This paper presents a new neural net architecture for decoupling objects from dynamics for the task of video prediction on simple scenes. Using several modifications to existing works, the authors encourage more decoupling of object features from other features like the position and dynamics of the inputs. The authors show improved accuracy on existing simple benchmark datasets over previous networks.


Minor typos (did not influence review):
Page 4: and is a hyperparameter of the model.
Page 9: we swap the positions swapped the blocks corresponding to shape


EDIT:
I can't seem to comment, so I'd like to add my comments here to the author's responses
Thank you for your updated wordings. I think the paper is clearer now, though I still think some images of the learning problems earlier in the paper during the problem definition portion would make it even stronger.

1. I see what you are saying, but I think of all the problems the only one that convinces me it could generalize to more complex scenes is CLEVR. The other experimental setups are too simple to be called "visual." I'd like to see more evidence on CLEVR and even harder datasets (maybe something with a physics engine and more realistic objects like Ai2THOR).

3. From the appendix: "We found that even though the permutation
module was trained on same time step object representations, it produces correct permutation matrix
even for 10 time step apart objects." How can you say it's correct if you don't have ground truth permutations?

**Strengths:**

+ The authors show an improvement on the existing state of the art on several benchmark datasets
+ The architecture accounts for several difficulties in training models to be disentangled in new/interesting ways
+ The paper is generally well-written

**Weaknesses:**

Clarity:
- I thought the explanation of the task and the actual problems came too late and were not depicted well enough for what the paper was trying to accomplish. 3 of the 4 datasets used I would call "toy" datasets of simple bouncing balls. The 3D dataset seems more visually driven, but even that uses CLEVR which is known to be visually simple. In the whole paper there is only a single picture depicting the actual task, and the tasks are only described in the experiment section. For a paper with "Visual Dynamics" in the title, I would have expected less toy problems, and more explanation to what the actual problems were. I think this could have been a stronger paper if it had foregone the visual component and worked directly with low level data.
- I found several explanations in the paper to be confusing/lacking detail. One key concept in the paper was that of a "Block". Here is the explanation from the paper: "Recently, (Locatello et al., 2020) in the paper on slot-attention proposed architecture for unsupervised discovery of objects via iterative refinement of what is referred to as a slot. Each slot binds to an object via attention over the entire image in their work. We extend their idea to the case of objects, where each slot now represents a block, which is iteratively refined by taking attention over a latent object representation" - My rephrasing of this is "rather than take attention over the entire image to get a representation, we first extract an object mask, and then take attention over that". I'm not sure that is correct, and even if it is, I don't understand why they need attention if they already have a mask.
- Another instance of this was the Permutation module. I did not understand the motivation behind it considering it is only used at training time. The authors say they learn a permutation to match up objects from one frame to the next, but that they supervise this permutation with ground truth knowledge. Then at test time, this component is removed. If you are already using ground truth information at train time, and at test time you don't use the module, why not just permute the features directly instead of learning a permutation matrix?
- There were a few other small questions I had about some other phrases. "All our object extractors are unsupervised and trained in a self- supervised manner. In our experiments, for 2D environments, we train an expert model similar to (Sharma et al., 2023b) to generate supervised data for Mask R-CNN." This seems to indicate the model is both unsupervised and self supervised, but then also trained with supervised data. That doesn't make any sense to me.

Experiments
- I thought the experiments in this paper were lacking in showing what the authors claimed. Predicting simple rigid body circle motion and even rigid body 3D synthetic CLEVR motion is not really convincing since it is such a problem removed from the complexities of the real world.
- To convince me that there is a decoupling happening, it is crucial to have an experiment that directly probes this decoupling. The ablation study in 4.4 seems to do that in some way, but I don't understand the experimental setup from the explanations, and again, it is only on a toy setup so I can't say whether it would generalize to more complex scenes. It sounds like somehow the authors took the embeddings from one color setup and swapped them with another color setup to look at the output. I guess it seems trivially obvious that the output should change color, but does that prove that the dynamics module only encodes dynamics or just that masking a part of the image results in color features for that part of the image.
- The DisFormer seems to be another reasonable ablation on paper but not carried out as well as I'd liked. Details are sparse, but it sounds like the entirety of MaskRCNN was replaced with an MLP, which doesn't seem like a reasonable substitution "by replacing the object extractor (refer Section 3) by an MLP to create dense object representations".

**Questions:**

I would like the authors to explain in more detail why the permutation model was necessary if it was only used during training. I would also like the authors to explain the DisFormer and experiment 4.4 better.

---

> ### Author Response · Authors · 2023-11-23
>
> 1. Thanks for pointing this out. We have added a task description in the introduction. While we agree that we have experimented only with visually simple datasets, we would like to to argue that disentanglement is a hard problem, and some of the SOTA techniques (Symbinder [1]) have only worked with images, which we have extended to video in our work. Further, SlotFormer [2] (which is a very recent work) for video prediction, also works on only simple rigid-body dynamics. We are working on compiling example images and hope to post them in the discussion phase.
>
> 2. We thank the reviewer for the suggestion and we agree with reviewers rephrasing. We would like to point out that we are not taking attention to the entire image and rather take attention to the image masked out by an object mask. We are aiming for disentangled representation of an object and thus we add that prior by inferring blocks which is iteratively refined by taking attention over latent object representation.
>
> 3. We want to highlight to the reviewer that the permutation gadget undergoes training in a self-supervised manner and we don’t have ground permutations. Its sole purpose is to align predicted object blocks with ground object blocks for dynamic loss computation. Importantly, during testing, there is no necessity for loss computation, and therefore, the permutation module is excluded from the testing process. We have added more details in the appendix.
>
> 4. We have included the specifics of object extractor training in the appendix. Additionally, it's worth noting that for the 2D environment, we employ Mask R-CNN as the object extractor. The training of Mask R-CNN necessitates supervised data, which is generated in unsupervised fashion from expert models.

---

> ### Author Response · Authors · 2023-11-23
>
> 1. We are unsure of the specific claim that the reviewer is referring to. To the best of our knowledge, we have not made any claim regarding our ability to do well with deformable objects. If the reviewer can point out a specific instance, we would be happy to correct it. While we agree that we have experimented only with rigid body surfaces, we would like to to argue that disentanglement is a hard problem, and some of the SOTA techniques (Symbinder refer) have only worked with images, which we have extended to video in our work. Further, SlotFormer [] (which is a very recent work) for video prediction, also works on only simple rigid-body dynamics. Hence, we strongly believe our setting is commensurate with some of the recent works in this area.
>
> 2. In response to the reviewer's suggestion, we have incorporated additional evaluation and experiments illustrating the decoupling of blocks in the appendix section F. Importantly, we emphasize that these blocks are learned in a fully unsupervised manner. One such instance is the block that has learned to represent color, as demonstrated in section 4.4. However, the final part of the review is currently unclear to us, and we would appreciate further clarification.
>
> 3. That was a typo and it should have been "by replacing the block extractor”. We have fixed it in the paper.

---

> ### Author Response · Authors · 2023-11-23
>
> We have addressed the need for a permutation model by including a specific example in the newly added content within the appendix section C. Furthermore, we have enhanced and clarified the details in section 4.4 based on your feedback.

---

### Meta-Review · Area_Chair_4mpy · 2023-12-01

**Metareview:**

This paper proposes an object-centric architecture for video prediction that disentangles information about motion dynamics from other information. This setting is quite interesting and novel, as pointed out by the reviewers, and the paper demonstrates how the proposed approach is able to improve over baselines. The method itself can be viewed as a novel combination of existing techniques.

Despite these strengths, there is broad agreement among the reviewers that the current submission is not ready for publication. In particular, several reviewers point out how the manuscript and the proposed method are difficult to understand (i.e. concerns about clarity), and at least two reviewers are concerned that the datasets considered are too simplistic to provide for a meaningful comparison. There are also some concerns that the proposed model is too complex and has not been sufficiently ablated, which leaves it unclear what components contribute most to the performance. Finally, more evidence is needed that disentanglement is indeed the reason for the observed improvement (and evaluating more difficult datasets might help with that). Overall, the author response falls short at addressing the majority of these concerns.

**Justification For Why Not Higher Score:**

* Concerns about clarity / presentation
* Substantial concerns about the experimental evaluation, including ablations, (slightly) more visually complex datasets, and overall limited evidence to  support the proposed working of the method and disentangling being the key to the reported performance.

**Justification For Why Not Lower Score:**

N/A

---

### Decision · Program_Chairs · 2024-01-16

Reject